# Electric Resistance of Elastic Strain Sensors—Fundamental Mechanisms and Experimental Validation

**DOI:** 10.3390/nano13121813

**Published:** 2023-06-06

**Authors:** Muchao Qu, Zixin Xie, Shuiyan Liu, Jinzhu Zhang, Siyao Peng, Zhitong Li, Cheng Lin, Fritjof Nilsson

**Affiliations:** 1School of Automobile and Transportation Engineering, Guangdong Polytechnic Normal University, Guangzhou 510450, China; dxqfz@outlook.com (Z.X.); siyaop171@gmail.com (S.P.); zt1123-@outlook.com (Z.L.); 13794476286@random.com (C.L.); 2Guangzhou Highteen Plastics Co., Ltd., Guangzhou 510800, China; dszzl@highteen.com.cn (S.L.); zhangjinzhunj@sina.com (J.Z.); 3KTH Royal Institute of Technology, School of Chemical Science and Engineering, Fibre and Polymer Technology, SE-100 44 Stockholm, Sweden; 4FSCN Research Centre, Mid Sweden University, SE-103 92 Sundsvall, Sweden

**Keywords:** strain-sensor, PDMS/CB, nanocomposite, modelling, electrical resistivity

## Abstract

Elastic strain sensor nanocomposites are emerging materials of high scientific and commercial interest. This study analyzes the major factors influencing the electrical behavior of elastic strain sensor nanocomposites. The sensor mechanisms were described for nanocomposites with conductive nanofillers, either dispersed inside the polymer matrix or coated onto the polymer surface. The purely geometrical contributions to the change in resistance were also assessed. The theoretical predictions indicated that maximum Gauge values are achieved for mixture composites with filler fractions slightly above the electrical percolation threshold, especially for nanocomposites with a very rapid conductivity increase around the threshold. PDMS/CB and PDMS/CNT mixture nanocomposites with 0–5.5 vol.% fillers were therefore manufactured and analyzed with resistivity measurements. In agreement with the predictions, the PDMS/CB with 2.0 vol.% CB gave very high Gauge values of around 20,000. The findings in this study will thus facilitate the development of highly optimized conductive polymer composites for strain sensor applications.

## 1. Background

Flexible strain sensors can be used in a wide range of applications, including flexible electronics, medical devices, health monitoring and VR-equipment [1,2,3,4,5]. The electrical resistance of most flexible strain sensor materials increases with increasing strain (*ε*), and this change in resistivity can be converted to electrical signals that are monitored and analyzed in motion analysis software programs. An ideal flexible strain sensor should have a wide operational strain range, a short response time, stable long-term properties, a linear strain response and high sensitivity (Gauge factor), i.e., a small deformation should produce a large change in the resistivity. However, it is difficult to optimize all of these properties simultaneously, so a good balance between the properties is required.

The development of flexible strain sensor materials is rapidly progressing, and a wide range of suitable composite material concepts have been proposed in the scientific literature. It is the case for all of these composites that electrically conductive materials are added into or onto a flexible, resistive polymer matrix, thus yielding an electrically conductive composite with strain-dependent resistance [6,7,8]. Examples of polymers that have been examined for strain sensor applications include elastomers (natural rubber [9,10,11], NBR [12,13], silicon rubber [14,15,16], PDMS [17,18,19] etc.), plastics (PET [20,21], Polyimide [22] et cetera), textile fibers (lycra [23,24,25], PUR [26], silk [27,28] etc.) and hydrogels [29,30,31,32]. Examined fillers include 0-dimensional “spheres” (carbon black [33,34,35], metal-oxide nanoparticles [36,37,38,39,40,41], quantum dots [42,43,44,45] et cetera), 1-dimensional “fibers” (carbon nanotubes [46,47,48,49,50], carbon nanofibers [51,52,53,54], carbon fibers [55,56,57]) and 2-dimensional “flakes” (graphene [58,59,60,61], graphene oxide [62,63,64,65], graphite nanoflakes etc.). 

Certain recent scientific studies on advanced functional composite materials and their application on strain sensor devices are worth reviewing even more carefully. Sun et al. [66] explored the use of strain sensor materials in the development of E-skin, a novel technology that enables real-time health monitoring and seamless integration with artificial intelligence systems. Another noteworthy study, by Sekine et al. [67], introduced a microporous-induced fully printed pressure-sensor, designed specifically for wearable soft-robotics machine-interfaces. The researchers developed composite materials to fabricate a sensor with enhanced sensitivity and durability, enabling precise detection and control of pressure changes. In the realm of integrated electronics, Sim et al. [68] presented a groundbreaking approach using high-effective-mobility, intrinsically stretchable semiconductors. Their fully rubbery integrated electronics demonstrated exceptional stretchability, allowing for seamless integration with curvilinear surfaces and conformal attachment to the human body.

Some recent research papers have also investigated the influence of mechanical external forces on the conductivity of various materials. Peng et al. [69] explored the material selection, design, and applications of nanocomposites in stretchable electronics. They demonstrated that the mechanical deformation of conductive polymer nanocomposites can induce changes in their conductivity, enabling the development of stretchable electronic devices with enhanced functionality and adaptability. Wang et al. [70] delved into the realm of natural biopolymer-based conductors for stretchable bioelectronics. They investigated how mechanical forces affect the electrical properties of biopolymer-based conductive materials and their suitability for biocompatible stretchable electronic applications. In a study by Kim et al. [71], material-based approaches for fabricating stretchable electronics were explored. The study emphasized the significance of considering the mechanical behavior of materials in the design and fabrication process. These recent studies collectively highlight the importance of investigating the impact of mechanical external forces on the conductivity of materials in the context of stretchable electronics. Understanding how materials respond to and adapt under mechanical deformation is crucial for the development of robust and reliable devices.

The idea of this paper arose after reading a large number of scientific papers on strain sensors, including those cited above. To our surprise, it is still very uncommon for scientific studies on strain sensor materials to analyze why the resistance changes with increasing strain in detail. The two dominating mechanisms are: (1) the resistivity of the material itself changes due to the strain; (2) the resistance increases when the sample becomes longer and thinner, i.e., geometrical effects. To the best of our knowledge, none of the previous research papers [1,2,3,4,5,6,7,8,9,10,11,12,13,14,15,16,17,18,19,20,21,22,23,24,25,26,27,28,29,30,31,32,33,34,35,36,37,38,39,40,41,42,43,44,45,46,47,48,49,50,51,52,53,54,55,56,57,58,59,60,61,62,63,64,65,66,67,68,69,70,71] have entirely accomplished the task of elucidating the two factors contributing to strain-induced changes in conductivity. As fundamental theoretical knowledge about conduction mechanisms is required when developing more robust and efficient strain sensor materials, this study aims to examine and explain the two resistance contributions and highlight how they interact. Additionally, the new theoretical knowledge was used to develop strain sensor composites (CB/PDMS and CNT/PDMS) with extremely high sensitivity (Gauge factor).

## 2. Methods 

### 2.1. Theory

Strain sensor materials are often characterized by their relative change in resistivity (*R*(*ε*) − *R*_0_)/*R*_0_ and their sensitivity, the Gauge factor *G*:(1)G(ε)=ⅆⅆεRε−R0R0≈Rε−R0R0ε
where *ε* = (*L − L_0_)*/*L*_0_ is the strain, *L* is the sample length (between the electrodes), *L*_0_ is the initial sample length, *R* is the electrical resistance and *R*_0_ = *R*(*ε* = 0) is the initial resistance. Furthermore, *v* is the Poisson’s ratio, *ρ_0_* = *ρ*(*ε = 0*) is the initial electrical resistivity and Δ*ρ* = *ρ*(*ε*) *− ρ_0_* is the change in resistivity due to the deformation *ε*. For small deformations, the Gauge factor can be approximated with the commonly used expression:(2)Gε≈1+2v+Δρ∕ρ0ε

However, flexible strain sensor materials are typically subject to large deformations, where Equation (2) is not applicable. More precise analytical expressions must therefore be derived in order to reveal how the relative change in resistivity and the Gauge factor is affected by the strain, as well as to distinguish between changes in the resistivity and purely geometrical effects.

Electrical resistance *R* is generally measured and computed using Ohms law:(3)$$R=\frac{U}{I}$$
where *U* is the (known) applied voltage and *I* is the measured current. With the electrical resistivity *ρ*(*ε*), sample length *L*(*ε*) and sample area *A*(*ε*), the resistance for a symmetric sample can be computed as:(4)$$R\left(\varepsilon \right)=\rho \left(\varepsilon \right)\frac{L\left(\varepsilon \right)}{A\left(\varepsilon \right)}$$

For many polymers, the electrical resistivity is nearly strain-independent, but for many polymer nanocomposites, the resistivity is strongly strain-dependent. One reason for this is that the filler particles become more separated in the stretching direction, but come closer in the transverse direction, as a consequence of the strain. Furthermore, high-aspect ratio filler particles, such as CNT, CNT and Graphene, reorient during stretching and gradually become more aligned with the direction of the applied force.

However, the resistance of a material sample will change with stretching, even if the electrical resistivity of the material is constant. The normalized length *L*/*L*_0_, volume *V*/*V*_0_ and area *A*/*A*_0_ of a symmetric material sample with Poisson’s ratio *v* can be computed as:(5)$$\frac{L\left(\varepsilon \right)}{L_{0}}=\left(1+\varepsilon \right)$$
(6)$$\frac{V\left(\varepsilon \right)}{V_{0}}={\left(1+\varepsilon \right)}^{\left(1-2v\right)}$$
(7)$$\frac{A\left(\varepsilon \right)}{A_{0}}=\frac{V\left(\varepsilon \right)∕V_{0}}{L\left(\varepsilon \right)∕L_{0}}={\left(1+\varepsilon \right)}^{\left(-2v\right)}$$

This gives the electrical resistance of the deformed sample:(8)$$R\left(\varepsilon \right)={\left(1+\varepsilon \right)}^{\left(1+2v\right)}\rho \left(\varepsilon \right)\frac{L_{0}}{A_{0}}={\left(1+\varepsilon \right)}^{\left(1+2v\right)}\frac{\rho \left(\varepsilon \right)}{\rho_{0}}R_{0}$$

In turn, the corresponding Gauge factor can be computed as:(9)Gε=ⅆⅆεRε−R0R0=1+2v1+ε2v+1+ε1+2v1ρ0ⅆρεⅆε
where:(10)1ρ0ⅆρεⅆε≈ρε−ρ0ρ0ε

If the resistivity is strain-independent, Equations (8) and (9) are simplified to:(11)$$R\left(\varepsilon \right)={\left(1+\varepsilon \right)}^{1+2v}R_{0}$$
(12)$$G\left(\varepsilon \right)=\left(1+2\nu \right){\left(1+\varepsilon \right)}^{2v}$$

For a typical rubber with *v* ≈ 0.5, the expressions are further simplified:(13)$$R\left(\varepsilon \right)={\left(1+\varepsilon \right)}^{2}R_{0}$$
(14)$$G\left(\varepsilon \right)=2\left(1+\varepsilon \right)$$

For such strain-independent rubbers, the Gauge values at 0% and 100% strain thus become 2 and 4, respectively.

The theoretically derived resistance equations can be compared with the corresponding semi-empirical equations from the literature. For instance, one such equation with two adjustable fitting parameters, *a* and *k*, was presented by Qu et al. [12]:(15)$$\frac{\Delta R\left(\varepsilon \right)}{R_{0}}=\frac{R\left(\varepsilon \right)-R_{0}}{R_{0}}=\left(1+k\varepsilon \right)\exp\left(a\varepsilon \right)-1\Rightarrow$$
(16)$$R\left(\varepsilon \right)=\left(1+k\varepsilon \right)\exp\left(a\varepsilon \right)R_{0}$$

If the resistances of Equations (8) and (16) are set to be equal, a semi-empirical expression for the normalized strain-dependent composite resistivity *ρ*(*ε*)/*ρ*_0_ can be derived:(17)$${\left(1+\varepsilon \right)}^{\left(1+2v\right)}\rho \left(\varepsilon \right)\frac{L_{0}}{A_{0}}=\left(1+k\varepsilon \right)\exp\left(a\varepsilon \right)R_{0}\Rightarrow$$
(18)$$\frac{\rho \left(\varepsilon \right)}{\rho_{0}}=\frac{\left(1+k\varepsilon \right)\exp\left(a\varepsilon \right)}{{\left(1+\varepsilon \right)}^{1+2v}}$$

The derivative of Equation (18), which occurs in the Gauge equation (Equation (9)), becomes:(19)1ρ0ⅆρεⅆε=exp⁡aε1+ε−2v+1akε2+akε+aε+a−2kvx+k−2v−1

Using these equations, the resistivity data from the literature and experiments will be analyzed.

### 2.2. Experimental

To assess and validate the theoretical models, experimental resistivity measurements were systematically performed on both carbon black (CB)/Polydimethylsiloxane (PDMS) nanocomposites and carbon nanotube (CNT)/PDMS nanocomposites, both with a wide range of filler fractions and strains. The composites were prepared by mixing the power (CB or CNTs) into the PDMS before crosslinking. The filler volume fractions ranged between 0 and 5.5 vol.% (in steps of 0.5 vol.%), and the filler was added and mixed into a large batch of liquid PDMS. After 30 min of ultrasonication, the curing agent was added, followed by mixing and vacuum removal of bubbles. The samples were poured into dog-bone tensile testing forms, which were put into an oven at 180 °C for 60 min for crosslinking. The prepared samples were stored in a desiccator until the electrical-mechanical measurements were performed. (Figure 1).

The electrical behavior of the composite specimen was determined using a Keithley 6487 Pico-ammeter connected to a tensile testing machine (Shenzhen SUNS Technology Stock Co. Ltd., Shenzhen, China, EUT5105), where the desired elongation of the samples could be provided. For each experimental condition, at least 5 membrane specimens were measured to ensure the reproducibility of the results.

## 3. Results and Discussion

### 3.1. Theory Results

The change in the electrical resistance with increasing strain can be partially attributed to geometrical effects and partly to strain dependent changes in the electrical resistivity of the material itself. The latter effect is more pronounced for composites than for pure polymers.

Composites with strain-dependent resistivity can be manufactured either through dispersing conductive particles into a resistive polymer matrix (“mixture composites”) or by coating conductive particles onto a resistive polymer sheet (“layered composites”). A large difference in the conductivity between the particles and the polymer is necessary to obtain a composite with strongly strain-dependent resistivity.

For mixture composites, the conductivity is low below the percolation threshold, where the conductive particles start to form conductive paths through the composite. To achieve the maximum strain dependence, the filler fraction should be slightly above the percolation threshold. The threshold depends on the size, shape, aspect ratio, dispersion and orientation of the conductive fillers. Small/thin particles with a high aspect ratio, such as graphene or CNT, provide the lowest threshold.

For layered composites, the particles in the conductive surface layer are initially (i.e., at zero strain) always in close contact, and most of them will continue to be in contact at larger strains. The maximum Gauge factor of layered composites is therefore (theoretically) lower than for mixture composites, but from a technical point of view, the coating process is an easier route to obtain strain-dependent resistivity, and a lower fraction of conductive particles is generally needed. However, the electrical properties of layered composites are generally more sensitive to abrasion and wear than mixture composites.

Note that plastic deformations of the material will lead to changes in the strain response pattern and increased energy dissipation, especially during the first stress-release cycle.

#### 3.1.1. Geometrical Effects and Comparison with Literature Data

The electrical resistance for a material sample with strain-independent resistivity increases with the sample length and decreases with the sample area (Equation (4)). When such a sample is stretched, the resistance in the strain direction will increase. However, how large is this effect compared to the contribution from the change in resistivity for a typical flexible strain sensor material? Figure 2 summarizes how the relative change in resistivity and the Gauge factor for a material with an intrinsically strain-independent resistivity is influenced by the strain and Poisson’s ratio. The data are calculated using Equations (8) and (9) with strains up to *ε =* 7, i.e., up to 700% elongation.

Both the relative change in resistivity and the Gauge factor increase with the Poisson’s ratio, with a maximum at *v* = 0.5. For perfectly strain-independent materials at 700% strain and *v* = 0.5, the maximum change in the relative resistivity is around 60 and the maximum Gauge factor is around 16. These geometrical effects are in addition to the eventual strain-dependent changes in the resistivity of the material itself.

Table 1 shows the Poisson’s ratios for some polymers that are commonly used in flexible strain sensors. Table 2 and Figure 3 show the maximum Gauge factors and strain ranges for strain sensor materials from the literature [9,10,11,14,20,22,27,72,73,74,75,76,77,78,79,80,81,82,83,84,85,86]. The comparison with Figure 2 shows that some of the real composites have Gauge factors in the same order of magnitude as the values achieved solely by geometrical factors. In future studies, it is therefore recommended to calculate and specify the contribution of the geometry for the relative change in resistivity and the Gauge factor, using Equations (8) and (9).

#### 3.1.2. Strain Dependent Resistivity of Mixture Composites

Many mixture composites have explicitly higher Gauge values than their geometrical contribution (Equation (9)), meaning that the resistivity of the composite itself has changed. Two mechanisms account for the change in resistivity: (1) during stretching, the particles become separated in the draw direction, leading to increased resistivity; (2) when composites with elongated fillers are stretched, the fillers become more parallel, which also affects the resistivity.

A particularly high Gauge value is anticipated when the filler fraction of the un-deformed sample is just above the electrical percolation threshold. When such a sample is stretched and the fillers become more separated, the resistivity of the stretched composite will eventually fall below the percolation threshold. This will lead to a huge increase in the resistivity and, consequently, in the Gauge number. Composites with nano-fillers, especially those with high effective aspect ratios, typically have low and sharp percolation thresholds, yielding high Gauge values. Composites with macro-fillers typically have higher and wider percolation thresholds, yielding lower Gauge values. With that knowledge, it is easier to design tailor-made flexible strain sensors with custom-chosen, strain-dependent Gauge numbers.

Figure 4 visualizes the deformations for three mixture composites that are stretched in the *x*-direction.

Figure 4a shows a composite with spherical fillers in an idealized foam (*v* = 0) before and after deformation. When the sample length is doubled, the particle concentration in the *y*- and *z*-directions remains constant, but decreases by a factor of two in the *x*-direction. If the resistivity in the *x*-direction of the non-deformed sample is *ρ_x_*(*ϕ*_0_*,ε =* 0), the resistivity at 100% strain is approximately *ρ_x_*(*ϕ*_0_*/2,ε =* 0), i.e., typically much lower than before the deformation. Ideal foams are thus expected to have high Gauge values. In real foams, however, the conductive particles will only reside in the solid phase of the foam, leading to somewhat lower Gauge values than anticipated.

Figure 4b also shows a composite with spherical fillers, but in a rubber matrix (*v* = 0.5). When this sample is stretched to double its initial length, the particle concentration in the *x*-direction is still reduced by a factor of two, but the concentration of the particles in the *y*- and *z*-directions increases. This should give a slower increase in the *x*-resistivity with the strain, i.e., lower Gauge values than for the corresponding foams.

Figure 4c shows a composite with elongated fillers, i.e., rods, such as CNT, or flakes, such as graphene. During the stretching, the particles gradually become more aligned to the strain direction, leading to a higher percolation threshold, but also to a higher maximum conductivity above the threshold, as conceptually sketched in Figure 5 [87]. This simultaneous change in the percolation threshold and maximum conductivity can be used to produce materials with a more linear response between the strain and resistivity. Note that the influence of the fiber reorientation during stretching is in addition to the other strain effects, e.g., that the fibers become more separated and that the sample becomes thinner and longer.

#### 3.1.3. Strain Dependent Resistivity of Layered Composites

Layered composites, with a conductive surface layer parallel to the strain direction coated onto the resistive polymer matrix, behave somewhat differently to mixture composites. Layered conductive composites are above the percolation threshold over a wide strain range, typically until microscopic- or macroscopic-strain-induced cracks form in the coating layer. A rough surface will lead to a slower strain response because the surface layer will unfold before it becomes stretched (Figure 6a). A thicker coating layer should thus, hypothetically, result in better long-term properties, as well as lower Gauge values (Figure 6b), whereas thinner coating layers are more sensitive to abrasion but have higher Gauge values (Figure 6c). The reason for this is that the particles become separated (through micro-cracks) more easily during stretching when the number of adjacent particles in the *y*-direction is smaller. Macroscopic cracks can also arise if the conductive layer is brittle, such that the particles do not easily glide and reform during the strain. Generally, lower Gauge values are anticipated for surface coatings and layered composites than for mixture composites.

The relative resistivity for two layered composites from the literature (NBR coated with either CB or CB/PDA) is plotted as a function of the strain in Figure 7. The rapid increase for such materials can be accurately modelled using Equation (15). The geometrical contribution to ΔR/R_0_, as computed by Equation (8), with strain-independent resistivity, is around 40 for the PDA/NBR/CB composite, i.e., roughly 1/3 of the total increase.

### 3.2. Experimental Results

Mixture nanocomposites comprising PDMS with different fractions of CB or CNT nanofillers were manufactured and analyzed at strains ranging between 0 and 40% to highlight the factors influencing the relative change in resistivity and the Gauge factor.

Scanning electron microscopy (SEM) was applied to the samples before stretching (Figure 8). The CB particles (Figure 8a) and the CNT particles (Figure 8b) are clearly distinguishable at 20,000 times magnification. The CB particles formed small clusters that were well separated from each other, whereas the CNT fillers were observed as randomly oriented white stripes.

The electrical conductivity (=1/resistivity) was measured for both the CB/PDMS and CNT/PDMS mixture composites at strains between 0% and 40%.

When the composite conductivity of the CB/PDMS composites was plotted against the particle filler fraction, a sigmoidal curve with a steep increase in conductivity around the percolation threshold was observed for the non-stretched samples (black dots). When the samples were stretched, the conductivity decreased, such that the curves were gradually shifted horizontally to the right (Figure 9). The effect of stretching is especially pronounced just above the percolation threshold of the non-stretched sample.

From the experimental conductivity data, the relative change in resistivity can be plotted versus the strain (Figure 10), showing modest values for 0.5, 3.0 and 5.5 vol.% CB, but an enormous increase at 2.0 vol.% CB, i.e., for filler fractions slightly above the percolation point of the un-stretched sample. This is not surprising as Figure 9 shows that the conductivity at 2.0 vol.% CB drops by three orders of magnitude after 40% strain. A comparison between Figure 7 and Figure 10 highlights the tremendously rapid increase in the relative resistivity with stretching for the 2.0 vol.% CB/PDMS composites. At first glance, Equation (16) does not seem to give a good fit to the data, but a closer examination of the raw data (from Figure 9) reveals that it is an effect of statistical errors from the experimental measurements. The reason for this is that small random deviations close to the percolation threshold have a large impact on the conductivity. When the data-points at 2.0 vol.% are instead extracted from the more reliable McLachlan curves [87] of Figure 9, a good fit is obtained with Equation (16) (inset of Figure 10). At strains higher than 40%, the curve will gradually reach a plateau when the resistivity of the composite approaches the resistivity of the polymer matrix.

The Gauge factors of these CB/PDMS composites (Figure 11) are computed with Equation (1), using the derivatives of the fitted curves from Figure 10. As expected, the Gauge values become very high for the 2.0 vol.% CB/PDMS composite, well above 20,000 at 40% strain. This is, to the best of our knowledge, the highest Gauge values reported for CB-based nanocomposites in the literature to date. The geometrical contribution from Equation (8) is negligible (0.96).

Corresponding experiments were also performed for the CNT/PDMS composites, as summarized in Figure 12, which shows electrical conductivity as a function of the CNT volume fraction at different strains. Interestingly, above the percolation threshold, the CNT/PDMS composites’ conductivity increases with the strain, which is opposite to the trend for the CB/PDMS composites. However, the experimental measurements agree with the theoretical forecasts, which predicted that the reorientation and alignment of the CNT fibers along the electrical field would result in increased conductivity at high filler fractions and a decrease at low filler fractions (Figure 5). Isotropic, randomly oriented fibers have a high probability to collide and form a conductive network, thus leading to a low percolation point. However, once percolation is reached, electron transport is more efficient when the conductive fibers are oriented along the electrical field, leading to higher conductivity for anisotropic fibers at high filler fractions.

## 4. Conclusions

The fundamental mathematical basis for the resistance–strain response behavior of flexible strain sensor materials was improved in this study by using a combination of theory and experiments. Equations were derived to compute how the relative change in the electrical resistance with strain is influenced by geometry and material-induced changes in the resistivity, respectively. The influence of geometry is often pronounced for flexible sensor materials with wide strain ranges. The strain-response mechanisms for layered composites (polymers coated with conductive particles) and mixture composites (polymers containing randomly distributed conductive particles) were thoroughly described for both spherical and elongated fillers. It was concluded that mixture composites at filler fractions just above the electrical percolation threshold of the unstretched sample will provide the maximum Gauge values. A very rapid increase in the conductivity around the threshold will potentially give higher Gauge values. To confirm this hypothesis, CB/PDMS and CNT/PDMS nanocomposites with 0–5.5 vol.% fillers were examined at strains between 0 and 40%. As expected, the composites with CB particles with filler fractions (2.0 vol.%) just above the percolation threshold gave very high Gauge values and a high relative change in resistance. At 40% strain, the Gauge value of the 2.0 vol.% CB/PDMS was above 20,000, which, to the best of our knowledge, is the highest Gauge value for CB-based strain sensor materials to be reported in the literature to date. An extensive compilation of strain-resistance response data from the literature was assessed and compared with our experimental data.

## Figures and Tables

**Figure 1 nanomaterials-13-01813-f001:**
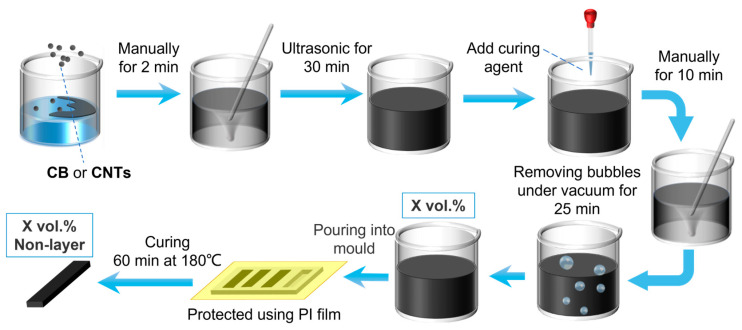
Procedure of making a PDMS-based (mixture) nanocomposite.

**Figure 2 nanomaterials-13-01813-f002:**
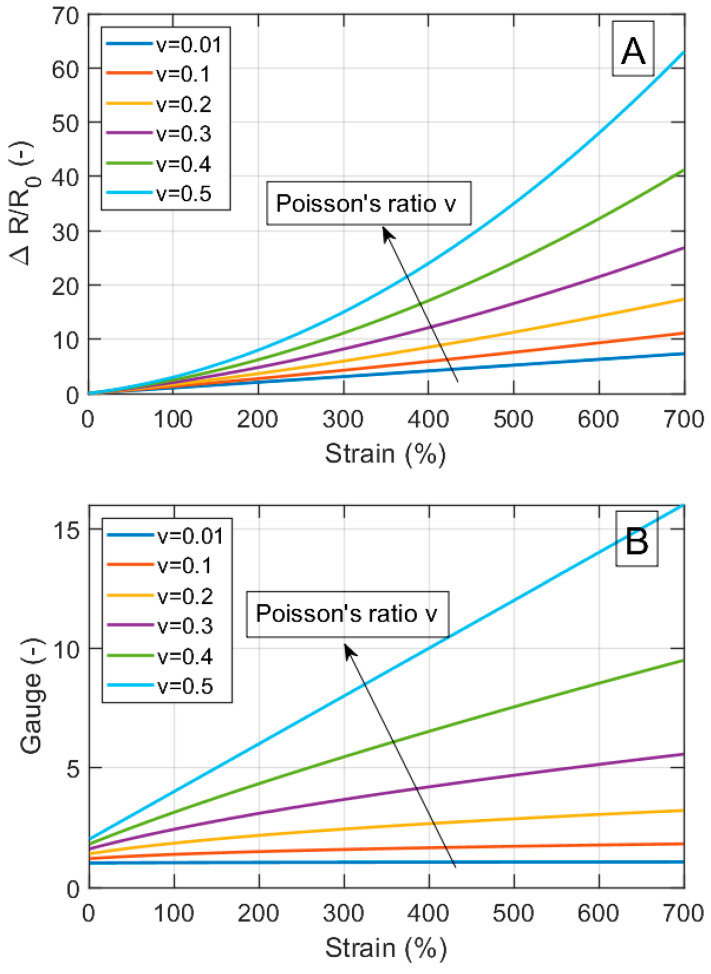
(**A**) Relative change in resistance (ΔR/R_0_) versus strain (ε) for different values of Poisson’s ratio (v). (**B**) Gauge (G) versus strain (ε) for different v.

**Figure 3 nanomaterials-13-01813-f003:**
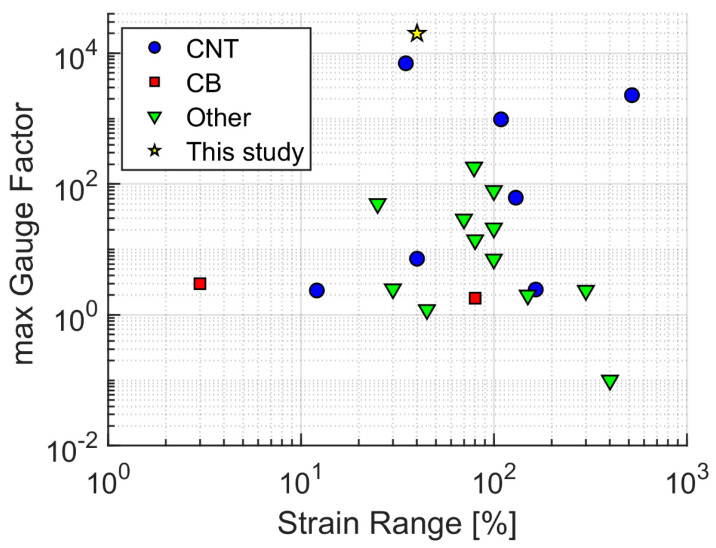
Maximum Gauge factor versus maximum strain range: Data from Table 1. The star shows the CB/PDMS composite with 2.0 vol.% CB at 40% strain from this study.

**Figure 4 nanomaterials-13-01813-f004:**
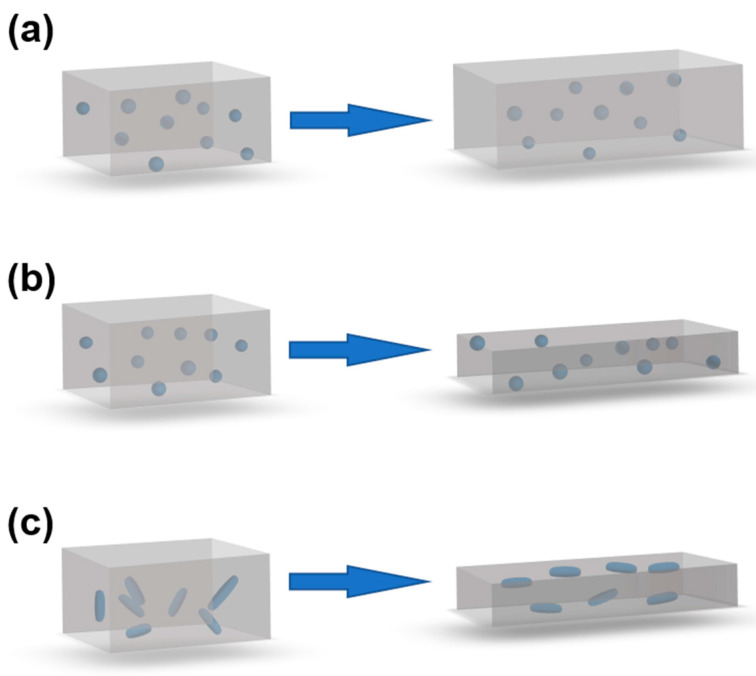
Left = non-deformed samples, right = deformed samples after stretching in the x-direction. (**a**) *v* = 0 (foam) with spherical fillers. (**b**) *v* = 0.5 (rubber) with spherical fillers. (**c**) *v* = 0.5 (rubber) with elongated fillers.

**Figure 5 nanomaterials-13-01813-f005:**
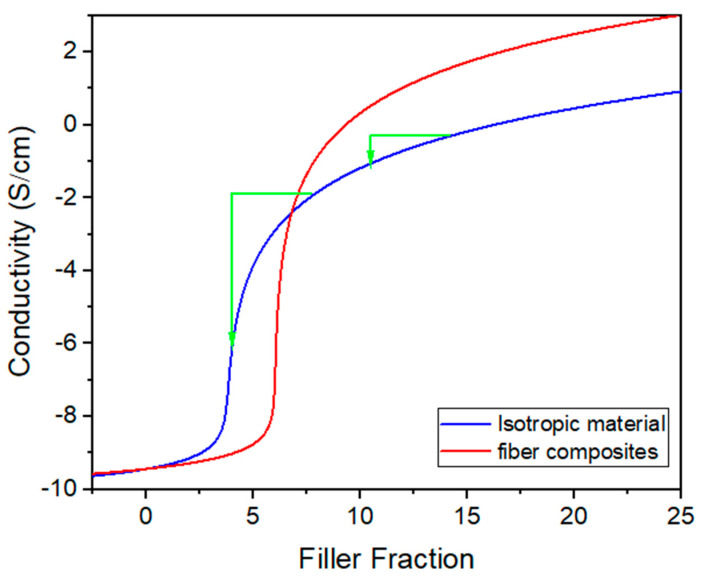
Electrical conductivity (=1/resistivity) as function of filler fraction for isotropic and anisotropic fiber composites with fibers oriented along the electrical field. The green arrows highlight that a small reduction in effective filler fraction (e.g. due to stretching) leads to a larger decrease in conductivity if the initial filler fraction is just slightly above the percolation threshold. This is a rough sketch drawn after [87].

**Figure 6 nanomaterials-13-01813-f006:**
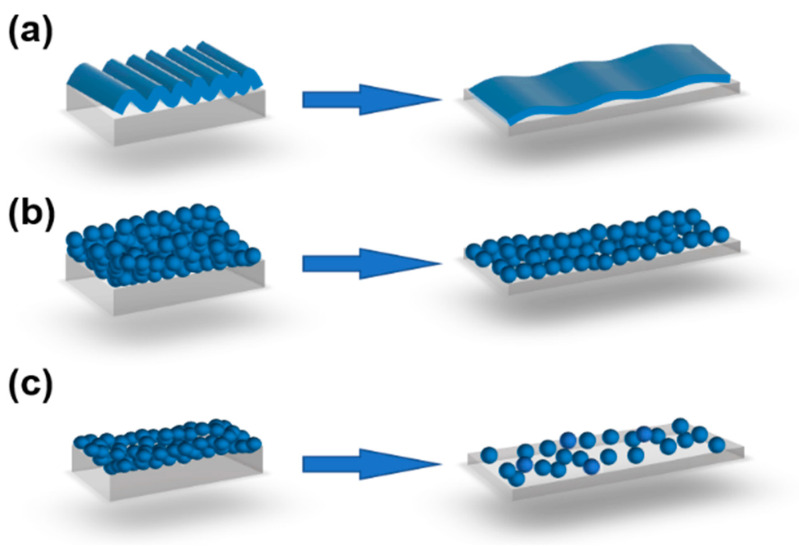
(**a**) A rough surface leads to a slow strain response because the wrinkles unfold before the individual particles are separated. (**b**) a thick coating layer enables a long stretching before individual particles are separated. (**c**) a thin coating layer leads to a strong stain response when the individual particles are separated.

**Figure 7 nanomaterials-13-01813-f007:**
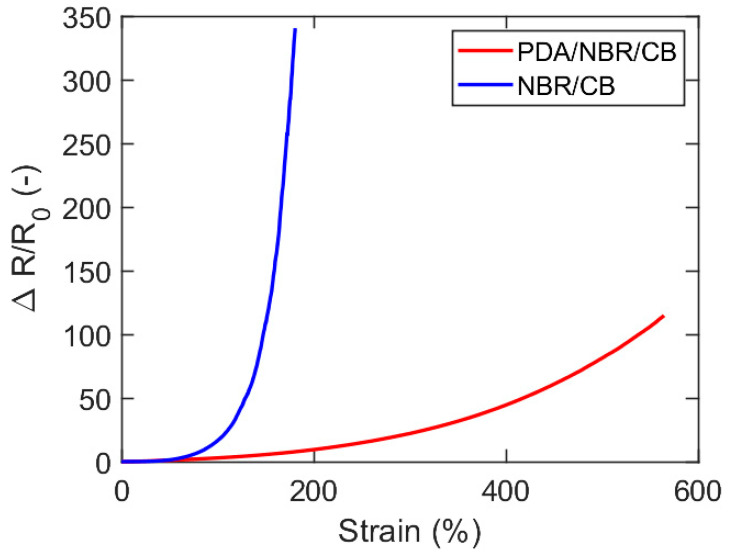
Relative change in resistivity as function of strain for layered composites comprising CB-coated NBR rubber. Drawn after [12].

**Figure 8 nanomaterials-13-01813-f008:**
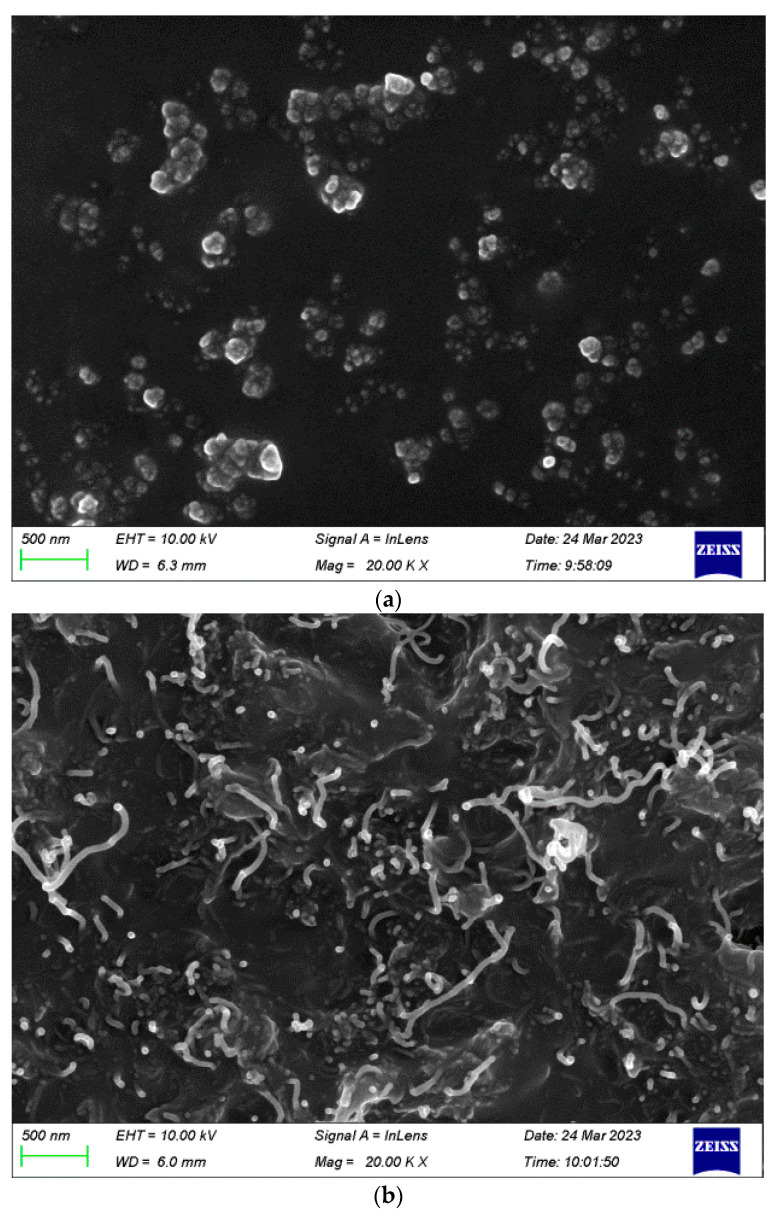
SEM micrographs of non-stretched nanocomposites at 20,000 times magnification. (**a**) CB/PDMS. (**b**) CNT/PDMS.

**Figure 9 nanomaterials-13-01813-f009:**
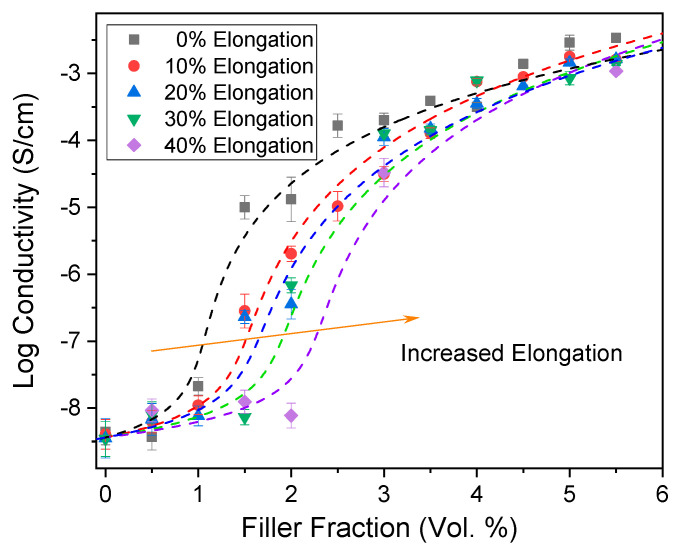
Electrical conductivity versus filler fraction for experimental CB/PDMS composites at different strain rates. The lines are fit with McLachlan’s equation [87].

**Figure 10 nanomaterials-13-01813-f010:**
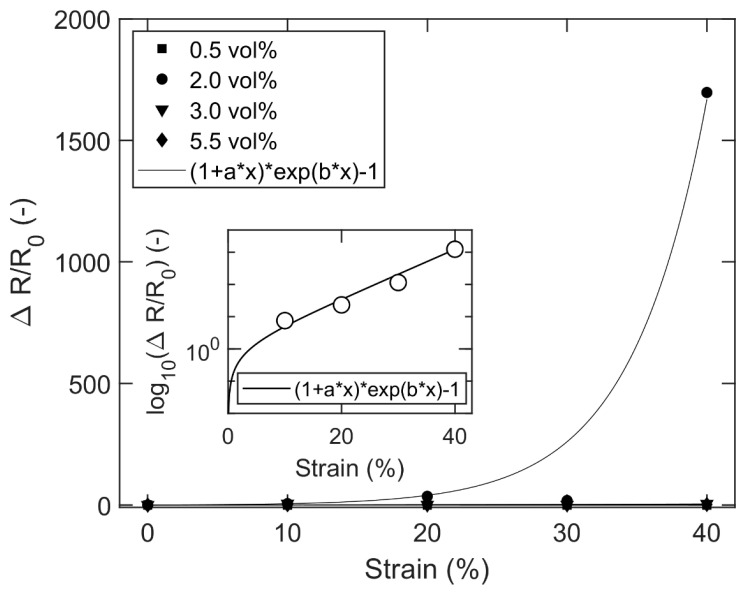
Relative change in resistivity (RCR) versus strain for mixture composites comprising PDMS with 0.5–5.5 vol.% CB. Inset figure: RCR versus strain for 2.0 vol.% composites, with data extracted from the McLachlan curves of Figure 9.

**Figure 11 nanomaterials-13-01813-f011:**
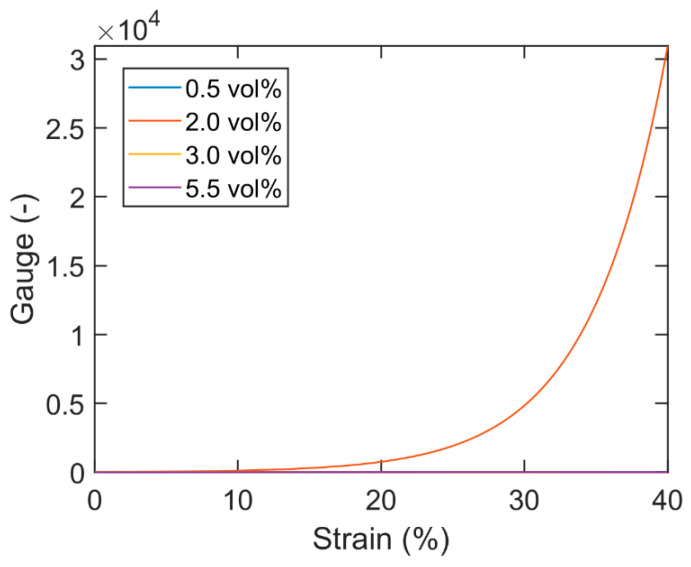
Gauge factor versus strain for mixture composites comprising PDMS with 0.5–5.5 vol.% CB. Note that the lines for 0.5, 3.0 and 5.5. vol.% are very small compared to the 2.0 vol.% curve and therefore overlap on this scale.

**Figure 12 nanomaterials-13-01813-f012:**
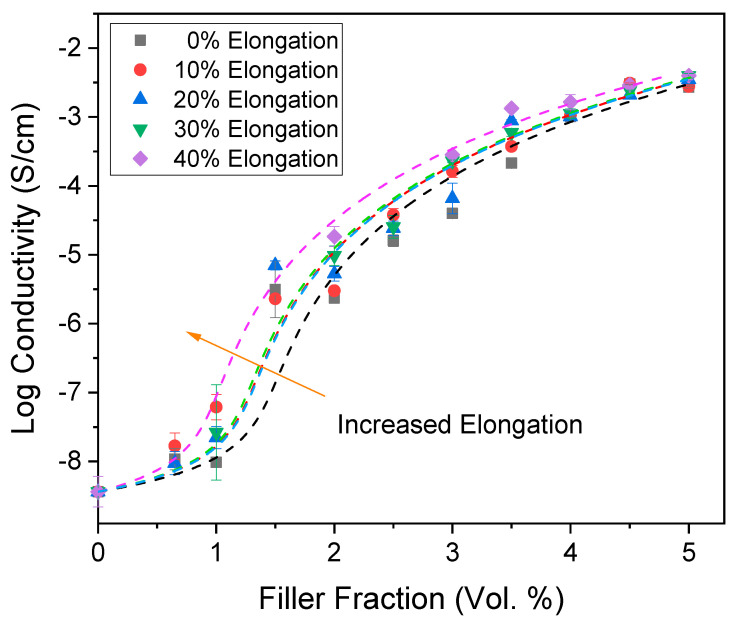
Electrical conductivity versus filler fraction for experimental CNT/PDMS composites at different strains. The lines are fit with McLachlan’s equation [87].

**Table 1 nanomaterials-13-01813-t001:** Poisson’s ratios for some polymers.

Resin	Poisson Ratios
PDMS	0.498
Polypropylene	0.43
TPU	0.3897
Natural rubber	0.48
PVC	0.38
Polycarbonate	0.36

**Table 2 nanomaterials-13-01813-t002:** Strange range and maximal Gauge factor of the composites in literature.

Resin	Filler	Strain Range	Max. GF	Reference
Polyimide/PDMS	Graphene	79	181	22
Silk	CNT	35	7000	27
Aerogel	CNT	12.1	2.36	72
PDMS	CB	80	1.8	73
Polypropylene	CB	3	3	74
PDMS	CNT	165	2.44	75
PDMS	MXene	45	1.2	76
PDMS	Graphene	30	2.49	77
TPU	CB/AgNWs	100	21.12	78
PDMS	MWCNTs	40	7.22	79
TPU	RGO	100	79	80
PDMS	CB/CNT	130	61.82	71
PU	PPy	300	2.36	82
PDMS	SACNT	400	0.1	83
PDMS	Graphene	150	2	84
PDMS	FGF	70	29	85
PDMS	Nanopaper	100	7.1	86
Polyester fabric	Ni@Fe₃O₄	40	-	20
Natural rubber	CNT	109	974.2	9
Natural rubber	CNT	520	2280	10
Natural rubber	MXene	80	14	11
Silicon Rubber	CCF	25	50	14
PDMS	CB (2.0 vol.%)	40	>20,000	This study

## Data Availability

All data in this manuscript are available by contacting the corresponding author.

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
