# Peer review of "Electric Resistance of Elastic Strain Sensors—Fundamental Mechanisms and Experimental Validation"

_nanomaterials, 2023, doi:10.3390/nano13121813_

Round 1

Reviewer 1 Report

Dear authors,

Please find my report in the attached file.

Kind regards.

Author Response

Comment 1: This is another interesting research work of the authors´ list about elastic strain sensors composed by a polymeric matrix and a conductive filler. The manuscript is well written and

organised including a proper number of references. Experimental results are clearly shown.

Response: Thank you for your kind words! Very appreciated!

Comment 2: However, I have a major concern about the explanation of Figures 9 and 10. Authors initially introduce their model given by Equation 16 [Ref. 12]. Nevertheless, they are not able to apply it to their experimental data for 2.0 % vol CB/PDMS and suggest a different model for R/R0 with new fitting parameters a, and b (see paragraph just below Fig. 9). This change of model is quite surprising with no further theoretical explanation nor fitting parameter extraction. This needs to be clarified.

Response: We see your point and after careful examination of the raw data and the theory we conclude that you are right. It is much better to fit the data with the reliable Eq. 16 than to introduce a new ad-hoc equation. In fact, after analysis of the data we can even see that the bad fitting with Eq. 16 is most probably just an effect of statistical errors from the experimental measurements. This can most easily be confirmed by also fitting 2.0 vol.% data points from the McLachlan fits of Fig. 9 with Eq 16. We have performed this assignment and presented the results in an inset of Fig. 10.  

Action: Figs 10, 11, 3 and table 1 are updated, and the following text is added in the results section: 

“A comparison between Fig 7 and Fig 10 highlights the tremendously rapid increase in relative resistivity with stretching for the 2.0 vol.% CB/PDMS composites. At a first glance, Eq. 16 does not seem to give a good fit to the data, but a closer examination of the raw data (from Fig. 9) reveals that it is an effect of statistical errors from the experimental measurements. The reason is that small random deviations close to the percolation threshold have large impact on the conductivity. When data-points at 2.0 vol.% are instead extracted from the more reliable McLachlan curves [ref] of Fig. 9, a good fit is obtained with Eq. 16 (inset of Fig. 10). At strains higher than 40 %, the curve will gradually reach a plateau when the resistivity of the composite approaches the resistivity of the polymer matrix.”

Figure 3. Maximum Gauge factor versus maximum strain range: Data from Table 1. The star shows the CB/PDMS composite with 2.0 vol. % CB at 40 % strain from this study.

Figure 10. Relative change in resistivity (RCR) versus strain for mixture composites comprising PDMS with 0.5-5.5 vol.% CB. Inset figure: RCR versus strain for 2.0 vol. % composites, with data extracted from the McLachlan curves of Fig. 9. 

Figure 11. Gauge factor versus strain for mixture composites comprising PDMS with 0.5-5.5 vol.% CB.

Comment 3: As a minor comment, I would suggest to use log y-axis for Figures 10 and 11 in order to observe better the experimental results and not only the spiky data.

Response: There are both pros and cons of presenting the data of Figs 10 and 11 in log-scale. Since we desire a simple comparison between Fig 10 and Fig 7 (which has linear y-scale), and since we want to highlight the enormous difference between the 2.0 vol curve and the others, we would prefer to keep the a linear scale on the y-axis. However, we have added an inset in Fig. 10 with log10 y-scale. We sincerely hope that this is ok with you. 

Action: An inset-figure with logarithmic y-axis is added to figure 10.

Reviewer 2 Report

In this study, the authors focused on “Electric Resistance of Elastic Strain Sensors – Fundamental Mechanisms and Experimental Validation”. Especially, they mentioned the main purpose of this study is elucidation the factors that contribute to the conductivity change of conductive composites. The experimental data are believed to represent factually. However, in my opinion, this article dose not reach the level of publication status. In particular, I think the description of the research contents and their background in the introduction is ambiguous.

In Introduction, the authors introduce conductive composite materials in REF9~65, but I think there is no meaning in these information. Here, you are just listing previous papers. You should explain how the authors came up with the idea for this work while comparison with previous research. In my understanding, the main purpose of this study is elucidation the factors that contribute to the conductivity change of conductive composites. In order for the readers to understand the process leading up to this goal, I recommend you modify structure of text as the following; (i) review the latest research on composite materials and there application of devices (in fact, the authors also mentioned device development in the introduction), (ii) it is necessary to introduce the latest research on the influence of mechanical external force on conductivity change, and (iii) the academic perspective of elucidating the factors of conductivity change that has not yet been achieved in the above latest research. When you follow the above advice, examples of the latest research is as the following:

(i) Adv. Mater. Technol., 8, 2201088. Adv. Intell. Syst., 2, 2000179. Sci. Adv., 5, eaav574.

(ii) ACS Appl. Mater. Interfaces, 13, 43831. Chem. Rev., 121, 2109. Adv. Mater., 32, 1902743.

Please proofread the English expression of the reconsidered sentence at the same time as the content of the introduction.

Author Response

Answer to reviewer 2

(1) In this study, the authors focused on “Electric Resistance of Elastic Strain Sensors – Fundamental Mechanisms and Experimental Validation”. Especially, they mentioned the main purpose of this study is elucidation the factors that contribute to the conductivity change of conductive composites. The experimental data are believed to represent factually. However, in my opinion, this article dose not reach the level of publication status. In particular, I think the description of the research contents and their background in the introduction is ambiguous.

--- Thank you for your kind suggestion. We have now added some more contents and rearranged the structure of the background. We sincerely hope the current version is satisfying and meets your requirements.   

(2) In Introduction, the authors introduce conductive composite materials in REF9~65, but I think there is no meaning in these information. Here, you are just listing previous papers. You should explain how the authors came up with the idea for this work while comparison with previous research.

--- Thank you for your kind suggestion. We have listed a huge amount of literature to check if the theme of our research has been investigated earlier and give a brief overview of the different types of composites that have previously been used for strain sensor applications. Even though all these papers focus on strain sensor materials, with all kinds of matrices and fillers, none of them has in sufficient detail discussed how strain-induces changes in sample geometry influences the electrical properties of strain sensors. The referenced papers are thus relevant to include just by confirming that most papers within the field of strain sensors does not analyze strain-induced geometry effects explicitly. 

We came up with the idea to this study when we were reading large amounts of scientific papers on strain sensor materials and noted that none of them explicitly distinguished between the two dominating mechanisms that influences the resistivity during stretching, i.e. macroscopic changes in the shape of the sample and microscopic changes in the material. This made us start investigating the two effects and we also stared to examine whether it is possible to use the theoretical understanding to construct composites with extremely high gauge and RCR values.  

Action: We have added the suggested description about how we came up with the idea of the study and how it is related to previous research. The following text is added to the introduction:

The idea of this paper came up after reading a large number of scientific papers on strain sensors, including those cited above. To our surprise, it is very uncommon that scientific studies on strain sensor materials in detail analyzes why the resistance changes with increasing strain. The two dominating mechanisms are (1) the resistivity of the material itself changes due to the strain (2) the resistance increases when the sample becomes longer and thinner, i.e. geometrical effects. To the best of our knowledge, none of the previous research papers [1-71] has fully accomplished the task of elucidating the two factors contributing to strain-induced changes in conductivity. Since fundamental theoretical knowledge about conduction mechanisms is required when developing more robust and efficient strain sensor materials, this study aims to examine and explain the two resistance contributions and highlight how they interact. Additionally, the new theoretical knowledge was used to develop strain sensor composites (CB/PDMS and CNT/PDMS) with extremely high sensitivity (gauge factor).

(3) In my understanding, the main purpose of this study is elucidation the factors that contribute to the conductivity change of conductive composites. In order for the readers to understand the process leading up to this goal, I recommend you modify structure of text as the following; (i) review the latest research on composite materials and there application of devices (in fact, the authors also mentioned device development in the introduction), (ii) it is necessary to introduce the latest research on the influence of mechanical external force on conductivity change, and (iii) the academic perspective of elucidating the factors of conductivity change that has not yet been achieved in the above latest research.

--- Sincerely thank you for your kind suggestion. This is indeed a great structure!

Action: We have now added the following contents to the background (in the introduction):

Certain recent scientific studies on advanced functional composite materials and their application on strain sensor devices are worth reviewing even more carefully. Sun et al. [66] explored the use of strain sensor materials in the development of E-skin, a novel technology that enables real-time health monitoring and seamless integration with artificial intelligence systems. Another noteworthy study by Sekine et al. [67] introduced a microporous-induced fully printed pressure-sensor, designed specifically for wearable soft-robotics machine-interfaces. The researchers developed composite materials to fabricate a sensor with enhanced sensitivity and durability, enabling precise detection and control of pressure changes. In the realm of integrated electronics, Sim et al. [68] presented a groundbreaking approach using high-effective-mobility intrinsically stretchable semiconductors. Their fully rubbery integrated electronics demonstrated exceptional stretchability, allowing for seamless integration with curvilinear surfaces and conformal attachment to the human body.

Some recent research papers have also investigated the influence of mechanical external forces on the conductivity of various materials. Peng et al. [69] explored the material selection, design, and applications of nanocomposites in stretchable electronics. They demonstrated that the mechanical deformation of conductive polymer nanocomposites can induce changes in their conductivity, enabling the development of stretchable electronic devices with enhanced functionality and adaptability. Wang et al. [70] delved into the realm of natural biopolymer-based conductors for stretchable bioelectronics. They investigated how mechanical forces affect the electrical properties of biopolymer-based conductive materials and their suitability for biocompatible stretchable electronic applications. In a study by Kim et al. [71], material-based approaches for fabricating stretchable electronics were explored. The study emphasized the significance of considering the mechanical behavior of materials in the design and fabrication process. These recent studies collectively highlight the importance of investigating the impact of mechanical external forces on the conductivity of materials in the context of stretchable electronics. Understanding how materials respond to and adapt under mechanical deformation is crucial for the development of robust and reliable devices.

(4) When you follow the above advice, examples of the latest research is as the following:

(i) Adv. Mater. Technol., 8, 2201088. Adv. Intell. Syst., 2, 2000179. Sci. Adv., 5, eaav574.

(ii) ACS Appl. Mater. Interfaces, 13, 43831. Chem. Rev., 121, 2109. Adv. Mater., 32, 1902743.

--- Sincerely thank you for your kind recommendation. We have cited all these six papers in a positive point of view, and in the corresponding position in the manuscript. The reference numbers and details are listed below:

  1. Sun, Q. J., Lai, Q. T., Tang, Z., Tang, X. G., Zhao, X. H., & Roy, V. A. (2023). Advanced Functional Composite Materials toward E‐Skin for Health Monitoring and Artificial Intelligence. Advanced Materials Technologies, 8(5), 2201088.
  2. Sekine, T., Abe, M., Muraki, K., Tachibana, S., Wang, Y. F., Hong, J., ... & Tokito, S. (2020). Microporous induced fully printed pressure sensor for wearable soft robotics machine interfaces. Advanced Intelligent Systems, 2(12), 2000179.
  3. Sim, K., Rao, Z., Kim, H. J., Thukral, A., Shim, H., & Yu, C. (2019). Fully rubbery integrated electronics from high effective mobility intrinsically stretchable semiconductors. Science advances, 5(2), eaav5749.
  4. Peng, S., Yu, Y., Wu, S., & Wang, C. H. (2021). Conductive polymer nanocomposites for stretchable electronics: material selection, design, and applications. ACS Applied Materials & Interfaces, 13(37), 43831-43854.
  5. Wang, C., Yokota, T., & Someya, T. (2021). Natural biopolymer-based biocompatible conductors for stretchable bioelectronics. Chemical Reviews, 121(4), 2109-2146.
  6. Kim, D. C., Shim, H. J., Lee, W., Koo, J. H., & Kim, D. H. (2020). Stretchable Electronics: Material‐Based Approaches for the Fabrication of Stretchable Electronics (Adv. Mater. 15/2020). Advanced Materials, 32(15), 2070118.

Round 2

Reviewer 1 Report

Reviser version has addressed all my suggestions.

Reviewer 2 Report

The authors responded sincerely to the comments. In particular, it became easier to understand the content of the Introduction. I have no further concerns.